# Influence of Alkyl Acrylate Nature on Rheological Properties of Polyacrylonitrile Terpolymers Solutions, Spinnability and Mechanical Characteristics of Fibers

**DOI:** 10.3390/ma16010107

**Published:** 2022-12-22

**Authors:** Ivan Yu. Skvortsov, Nikita M. Maksimov, Mikhail S. Kuzin, Roman V. Toms, Lydia A. Varfolomeeva, Elena V. Chernikova, Valery G. Kulichikhin

**Affiliations:** 1Institute of Petrochemical Synthesis Named after A.V. Topchiev, Russian Academy of Sciences, Leninsky Ave., 29, 119991 Moscow, Russia; 2Faculty of Materials Sciences, Lomonosov Moscow State University, Leninskiye Gory, 1, Building 73, 119991 Moscow, Russia; 3Institute of Fine Chemical Technologies Named after M.V. Lomonosov, MIREA–Russian Technological University, Vernadsky Ave., 86, 119571 Moscow, Russia; 4Faculty of Chemistry, Lomonosov Moscow State University, Leninskiye Gory, 1, Building 3, 119991 Moscow, Russia

**Keywords:** polyacrylonitrile, reversible addition-fragmentation chain-transfer polymerization, comonomer sequence, rheology, fiber spinning, alkyl acrylate, acrylamide

## Abstract

The influence of alkyl acrylate comonomers in the rank of methyl- (MA), butyl- (BA), ethylhexyl- (EGA), and lauryl- (LA) in ternary copolymers based on acrylonitrile, alkyl acrylate and acrylamide (PAN-alkyl acrylate) on their solutions rheological behavior in dimethyl sulfoxide (DMSO), and mechanical properties of the spun fibers have been investigated. To reveal the role of molecular weight, two series of copolymers with molecular weights of ~50 and 150 kg/mol have been studied. It was shown that the nature of the alkyl acrylate does not significantly affect the rheological behavior of their solutions regardless of the length of the alkyl substituent and the content of the alkyl acrylate in copolymers. An exception is the high-molecular PAN-LA, which is characterized by a non-Newtonian behavior at lower concentrations. Two series of fibers were spun from the characterized ranks of low and high-molecular-weight copolymer solutions. For all copolymers, a 2.5–5-fold increase in the strength and elastic modulus of the fiber was found with an increase in *M_w_*. It has been shown that PAN-MA and PAN-LA fibers have a tensile strength of 800 MPa that is 1.5–3 times higher than that of other copolymers spun in the same conditions.

## 1. Introduction

Acrylonitrile actively interacts with acrylates and carboxylates which leads to the probability to obtain a variety of semi-crystalline, flexible-chain copolymers with good spinnability [1,2]. The fibers and films obtained from them are characterized by high mechanical characteristics and are used in various high-tech areas such as the textile industry and some segments of special techniques, since the fibers based on acrylonitrile (AN) copolymers are the main precursors of carbon fibers [3,4,5,6]. The melting point of PAN exceeds the temperature of its cyclization [7], which makes it impossible to process it into products via the melt. Thus, routine technological processes for obtaining final fibers are based on a solution technology [8,9,10,11]. For the PAN structure, the strong nitrile–nitrile interactions are intrinsic, so its solutions are prone to structuring (gelling), which reduces their stability over time and complicates processing. Therefore, it is important to choose the optimal solvent [8,12,13]. The evaluation of the polymer–solvent interaction in a wide range of concentrations was carried out by rheological methods [14,15,16], which give food for understanding some processes occurring in polymer solutions. Proving these ideas with FTIR methods allows ones to enter deeper into these structural features of PAN solutions in different solvents [12].

Traditionally, the synthesis of PAN is carried out by anionic [12,17,18,19,20] and radical polymerization [21,22]. Anionic polymerization is characterized by high values of the chain transfer constant [23] and is mainly used to obtain branched polymers with narrow molecular weight distribution (MWD) [24]. For the synthesis of a linear copolymer suitable for obtaining high-quality fibers [25], the radical polymerization method, including the reversible addition-fragmentation chain transfer (RAFT) process, has proven itself well, which allows obtaining a narrower MWD and a uniform distribution of comonomer units [26,27,28,29,30,31,32].

Comonomers are usually introduced in the homopolymer to reduce its degree of crystallinity [33], improve solubility, and increase flexibility, which is necessary to improve fiber spinnability [34], and to control thermal behavior during the oxidation and carbonization stages in the case of carbon fibers preparation [35,36,37,38,39]. The most commonly used are binary and ternary copolymers with alkyl acrylates and monomers containing carboxyl or carboxylate groups: acrylic [40,41], itaconic [42,43], methacrylic [44,45] and other acids, acrylamide [46,47], etc.

Alkyl acrylate comonomers are used as internal plasticizers to reduce intermolecular and intramolecular interactions of nitrile groups [48]. Most often, methyl acrylate is used [49,50,51]; however, other compounds such as butyl acrylate, ethylhexyl acrylate, lauryl acrylate, etc., can also be used, but much less attention has been paid to them in modern literature. It is worth noting the work [52], where the thermal behavior of PAN copolymers with various alkyl acrylates was studied, and it was shown that the activation energy of nitrile groups cyclization, regardless of the nature of the alkyl acrylate comonomer, remains constant, about 100 kJ/mol.

In [53], data are presented on the synthesis of copolymers of AN with acrylamide with both narrow and bimodal MWD distributions. It is shown that the viscosity of their solutions is lower than for traditional free-radical synthesis at the same MWD. The differences in the branching degree between AN and acrylamide copolymers could be a reason for that and also leads to various thermal behavior of the copolymers at low temperatures. In continuation of this work [54], it was shown that ionic cyclization in terpolymers with acrylamide develops faster and more completely than in terpolymers of similar composition with acrylic acid. Therefore, having studied the thermal behavior, we moved on to solutions and fiber spinning.

In this work, we tried to answer some questions about the choice and use of various alkyl acrylate comonomers, in particular, the effect of their alkyl chain length and degree of branching in PAN terpolymers with acrylamide of various molecular weights on solubility, viscoelastic characteristics of solutions, the spinnability and fiber properties obtained from them.

## 2. Materials and Methods

### 2.1. Materials

The synthesis of acrylonitrile terpolymers with acrylamide and various alkyl acrylates (methyl-, butyl-, 2-ethylhexyl-, and lauryl-) was carried out by the RAFT method and described in detail in [54]. Two series of terpolymers AlkA1 and AlkA2 (where Alk is the corresponding alkyl acrylate) with *M_w_*~150 (AlkA1) and ~50 kg/mol (AlkA2), respectively, have been synthesized. The composition and characteristics of the terpolymers are presented in Table 1 and Table 2.

Before dissolution in a solvent, the terpolymer was dried for 4 h at 80 °C using a vacuum evaporator Heidolph Hei-vap produced by Heidolph Instruments (Schwabach, Germany). DMSO (99.6% produced by EKOS-1, (Moscow, Russia) was dried using A4 molecular sieves (Sigma Aldrich, St. Louis, MO, USA) [12], and the residual moisture was determined by coulometric titration using the instrument Expert 007M (produced by Econix Expert Ltd., Moscow, Russia). The residual moisture content in all solvents did not exceed 0.05%.

The viscometric characteristics of the solutions were described earlier [54] and some of them are presented in Table 2 for a better understanding rheological behavior of solutions in different concentration ranges. The concentration of entanglement formation was determined from the intersection of the linear dependences of the specific viscosity on the reduced concentration *c*[*η*], which characterizes the volume occupied by the polymer coil in solution and describes the behavior of solutions in dilute and concentrated ranges. The transition concentrations from dilute to a semi-dilute solution (*c**) and from a semi-dilute to concentrated entangled (*c***) solution were determined by the beginning of the deviation from linear dependences of the specific viscosity of terpolymer solutions in DMSO at 25 °C on the dimensionless parameter *c*[*η*], as shown in [54].

To study the rheological properties, solutions in a range of terpolymer concentrations from 3 to 40% were prepared. The spinning solution’s concentrations varied from 30 to 45%, depending on the polymer molecular weight. Solutions were prepared in sealed vials using a laboratory-made J-shaped anchor rotor mixer with a rotor speed of 10 rpm, for at least 10 h at 70 °C.

To exclude the gelation effect of concentrated solutions at room temperature, which is typical for PAN solutions in various solvents [12,55], all solutions were kept at 70 °C for 4 h before experiments.

### 2.2. Methods

#### 2.2.1. Rheological Measurements

The systematic rheological investigation of all solutions has been performed using the rotational rheometer HAAKE MARS 60 (Thermo Fisher Scientific, Karlsruhe, Germany). Experimental data were obtained using the “cone-plate” geometry of the measuring unit with diameters of 20 and 60 mm, and the angle between conical and plate surfaces equal to 1 deg.

A protective cup was installed during the experiment to prevent moisture absorption from the air. Moreover, the sides of the measuring unit were covered with polydimethylsiloxane liquid, which prevents solvent evaporation and any contact with air humidity.

Flow curves for all solutions under study were measured at a steady-state regime of shearing in the range of shear rates from 0.1 to 1000 s^−1^.

The frequency dependencies of the complex modulus of elasticity components storage modulus and loss modulus were measured in the frequency range of 0.628−628 rad/s in the linear domain of the viscoelastic behavior.

At least 2 of the same solutions were prepared, and 3 and more repeats were made for each concentration during experiments for the repeatability.

#### 2.2.2. Fiber Spinning

Fibers were spun on the laboratory-made spinning line using a mechanotropic method, which does not require the use of coagulation baths, while phase separation occurs mainly due to high tensile strain ratios at the spinning drawing stage of solution jets [56]. Since the jet of the solution is exposed to air for a long time upon flowing out of the capillary and until the formation of the gel fiber, its relative humidity was controlled and is not exceeded ~20%. An aqueous wash bath was used to clean the fiber surface from the residual solvent. The scheme of the laboratory mechanotropic spinning line is shown in Figure 1.

#### 2.2.3. Fiber Properties

Fiber morphology and diameter measurements were studied by optical microscopy using a Biomed 6PO (Moscow, Russia) combined with a ToupTek XFCAM1080PHD camera (ToupTek, Hangzhou, China).

The analysis of mechanical properties was carried out on samples with a length of 1 cm at 25 ± 2 °C using an Instron 1120 tensile testing machine (Norwood, MA, USA). The results were averaged over at least 10 tests performed.

The study of thin sections of fiber obtained by the mechanotropic method was carried out on a transmission electron microscope LEO 912 ab omega (LEO Carl Zeiss SMT Ltd., Jena, Germany) at an accelerating voltage of 100 kV. To study the morphology of fiber cross-sections, the samples were prepared as follows. A bundle of fibers was poured with an epoxy resin to obtain a microplastic in the form of a cylinder with a diameter of ~500 μm, from which a cut was made perpendicular to the long axes of the fibers on an Ultracut-R Ultramicrotome (Leica Microsystems, Wetzlar, Germany).

## 3. Results and Discussion

### 3.1. Rheology

Previously, in [54], it was shown that variation of the alkyl acrylate comonomer in a ternary copolymer has almost no effect on the affinity of the polymer for DMSO and does not affect the rheological properties of solutions in the range of low strain rates corresponding to the maximal Newtonian viscosity. During the spinning process, the solution is subjected to deformations over a wide range of shear rates, that necessity for a more detailed study of the rheological behavior.

The flow curves of semi-dilute and concentrated solutions of a series of copolymers with *M_w_*~150 kg/mol (AlkA1 rank) are shown in Figure 2.

For all solutions from the copolymers under consideration, the expected proportional increase in viscosity with increasing concentration is observed. For a series of high molecular weight copolymers, an increase in concentration from 3 to 30% leads to an increase in viscosity by four orders of magnitude, and the region of the plateau of the highest Newtonian viscosity narrows from 10^4^ to 0.1 s^–1^ for 3% and 30% solutions, respectively.

For spinning fibers, the concentrated solutions with the maximal Newtonian viscosity of about 10^4^ Pa·s are usually applied. Figure 3 compares in more detail the flow curves of such highly viscous systems for different copolymers at 25 and 70 °C.

It is seen that the behavior of solutions of all copolymers with the high molecular weight AlkA1 series, except for PAN-LA1, is similar and is determined solely by the molecular weight of the polymer and its concentration in the solution. The presence of a long alkyl acrylate affects the flow regime of high-molecular-weight systems: with a comparable viscosity, PAN-LA1 solutions are characterized by a non-Newtonian behavior at much lower shear rates, especially at 25 °C.

The behavior of concentrated low molecular weight samples (Figure 3c,d) with a viscosity similar to solutions of high molecular weight polymers is similar for all systems, including the lauryl acrylate copolymer. In this case, even for highly concentrated solutions (45%), a fairly wide plateau of the highest viscosity is observed. It means a smooth change of viscosity with shear rates.

For a deeper understanding of the alkyl acrylate comonomers’ role in the behavior of terpolymer solutions, their viscoelastic properties were studied in a wide range of concentrations. The frequency dependences are shown in Figure A1 and Figure A2 in the Appendix A. Figure 4 shows the dependences of the exponents of the frequency dependences of the elastic and loss moduli in logarithmic coordinates for solutions in the low-frequency zone corresponding to the maximum chain relaxation times. The obtained data were normalized through the parameter *c*[*η*], which characterizes the excluded volume or the fraction occupied by polymer coils in solution, which made it possible to take into account the differences in the molecular weight of the samples to reveal the effect of the alkyl acrylate nature.

It can be seen that in the region of concentrated solutions (marked with a vertical dotted line), the behavior of solutions starts to deviate significantly from the model of a standard viscoelastic fluid in the low-frequency region, which obeys the Maxwellian behavior with slope angles in the terminal zone of 1 and 2 for the loss and elasticity moduli, respectively. This is due to the increase in relaxation times in concentrated solutions. It should be noted here that the role of the alkyl acrylate does not significantly affect the behavior of the viscous component of the complex modulus, while the elastic behavior of PAN-LA series solutions does not obey the general dependence, due to the limited solubility of that polymer in DMSO at 25 °C.

A sharper decrease in the slope of the elastic modulus with increasing concentration is apparently due to the influence of the long lauryl acrylate fragments, which increases intermacromolecular interactions and leads to some difficulty with its dissolution. The matter is that all copolymers at room temperature, except for PAN-LA1, form homogeneous solutions over the entire range of concentrations under study. However, PAN-LA1 under these conditions is only partly soluble in DMSO up to ~20 wt%. Higher concentrations are stable dispersions containing gel-like swollen polymer fragments (Figure 5b). The content of such fragments increases with increasing concentration and decreases with increasing temperature; therefore, PAN-LA1 forms a homogeneous 35% solution only when heated above 70 °C (Figure 5a). We can also observe the reverse process when a clear solution begins to become cloudy when cooled to 25 °C a few hours after heating (Figure 5c).

On the frequency dependences of the complex modulus components, in the region of transition to the rubber-like state, which is detected upon reaching the maximum of loss modulus, the role of the alkyl acrylate comonomer does not appear at all. This is most clearly observed by changing the crossover frequency, i.e., equalities *G*′ and *G*″ for solutions of different concentrations. To exclude the influence of the molecular weight of the polymer, the data were normalized through the parameter *c*/*c*^#^ characterizing the distance from the critical concentration of the beginning of the entanglement’s formation (*c*^#^) (Figure 6).

At high frequencies (above the crossover frequency), the solution behaves like a cross-linked gel, and its properties are determined solely by the density of the entanglements that have arisen in these conditions. The forerunners of them are fluctuating entanglements, which can be considered as a precursor of rubber-like nodes. As the mesh density increases, which can be characterized by the *c*/*c*^#^ ratio, the crossover frequency decreases linearly in semilogarithmic coordinates.

Thus, among the studied alkyl acrylates, only lauryl acrylate comonomer introduces a certain specificity into the behavior of solutions, which is associated with the difficulty of dissolving PAN-LA, causing the presence of heterogeneity in solutions.

### 3.2. Fiber Spinning

Based on the data obtained, concentrations of the copolymer solutions with low and high molecular weights (indicated in Table 3) with similar viscoelastic properties were selected. This made it possible to spin fibers under similar viscous and temperature conditions, which is especially important for PAN-LA1 solutions. During fiber spinning from polymer solutions, one of the most important parameters that determine the mechanical properties is the orientation stretching [57]. Preliminary experiments showed differences in the solution jet behavior under tension; therefore, the main task of this stage was to obtain fibers with the maximum stretching ratio. To do this, the maximum rate of stable spinning was successively chosen and measured at each stage, which in the case of the mechanotropic process mainly determines the spin-draw ratio. Next, plasticizing drawing was performed in the air, washing from the residual solvent, drying, and thermal stretching at a temperature (100 °C) above the glass transition temperature stages. The used spinning regimes are summarized in Table 3.

The surface morphology of fibers formed in similar spinning conditions is shown in Figure 7. Images were obtained by optical microscopy in transmission light.

It can be seen that all fibers are transparent and do not have surface finger-like defects caused by the action of the washing agent. The larger diameter for PAN-LA2 fibers is explained by the lower degree of stretching (Table 3).

Cross-section images of the PAN-MA1 sample obtained by the TEM method are shown in Figure 8.

The spun fibers are characterized by a round cross-sectional shape (the observed oval shape is due to the not exactly perpendicular direction of the fiber bundle cut by the microtome), and typical defects which are often found in fibers spun by wet or dry-wet processes using a coagulant [58,59] are absent.

### 3.3. Mechanical Properties

The diameters and mechanical characteristics of the obtained fibers for a series of PAN terpolymers of various compositions with high (1) and low (2) molecular weights, obtained at maximum draw ratio, are presented in the form of histograms in Figure 9.

In both series with similar molecular weights of copolymers, the fibers’ strength with methyl acrylate significantly exceeds the strength of samples with other alkyl acrylates. In the low-molecular series, the strength of PAN-MA2 samples is about 3 times, and the elastic modulus is 1.5–4 times higher than that of fibers obtained from solutions of other copolymers, while the relative elongation at break values is quite high (about 20%), that could indicate the best macromolecular orientation of methyl acrylate containing copolymer during spinning.

In the high-molecular series, the differences are not so significant: the strength of PAN-MA1 increases by 1.5–2 times, and the elastic modulus increases by 10–30% compared with other copolymers at the same values of the relative elongation. However, the absolute values of these parameters are significantly higher for this copolymer. An increase in the molecular weight of the polymer has a positive effect on the mechanical properties of the fibers for all copolymers: the strength increases by 2.5–6 times, and the elastic modulus by 1.5–4 times with an increase in *M_w_* from ~40 to ~175 kg/mol (Figure 10). In the saturation zone, the maximal values of mechanical characteristics are observed for MA1.

The role of the drawing ratio is extremely important for the mechanical characteristics of fibers. As a criterion reflecting the stretching value, the fiber diameter was chosen, which decreases as the filament is stretched. The relevant data is shown in Figure 11.

For all studied PAN compositions, the strength and modulus increase with a decrease in the diameter which means an increase in the draw ratio. Among the copolymers with various alkyl acrylates, the PAN-MA1 copolymer is the best. The closest to the strength parameters of the PAN-MA1 terpolymer are fibers from the high-molecular-weight copolymer PAN-LA1 (the samples are marked with an arrow in Figure 11a,b). Probably, this result is due to the implementation of a high thermal stretching ratio of fibers from these copolymers. For PAN-BA and PAN-EHA terpolymers, it was not possible to realize similar drawing conditions for obtaining the high-strength samples.

As for the deformability of the fibers, characterized by elongation at break, it decreases with decreasing diameter, i.e., the higher the draw ratio, the stiffer the fibers become. However, the retention of relative elongation values at diameters of 10–20 µm in the range of 20–25% indicates the possibility of additional orientation at the stage of fiber thermolysis upon heating above the glass point and up to the onset of cyclization and the formation of conjugated sequences.

## 4. Conclusions

The nature of alkyl acrylate has practically no effect on the rheological properties of PAN terpolymers, except for concentrated solutions of lauryl acrylate-containing terpolymers, which are limitedly soluble in DMSO up to the temperature of 70 °C. The maximal draw ratio during mechanotropic spinning of fibers from various copolymers differs, but in the optimal regimes for each sample, both low-molecular and high-molecular polymers form uniform fibers with a round cross-section.

For all studied terpolymers, a clear correlation between mechanical properties and molecular weight (at approximately the same MWD) was observed. It has a decisive influence on the mechanical properties of the obtained fibers.

The nature of the alkyl acrylate also has a significant effect on the mechanical properties of the fibers. Among the used alkyl acrylate comonomers, MA and LA have the best properties. Their strength was 1.5–6.0 times higher, and the elastic modulus was 1.1–4.0 times higher than for the other copolymers. The detailed selection of thermal drawing conditions allows obtaining fibers from high-molecular-weight terpolymers, containing MA and LA, with a strength of up to 900 MPa and elastic modulus close to 10 GPa at relative elongation at a break of ~20%.

## Figures and Tables

**Figure 1 materials-16-00107-f001:**
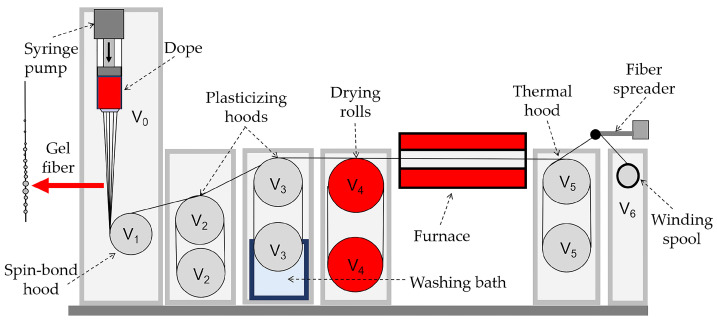
Scheme of the mechanotropic spinning line. *V*_0_ is the speed of the outflow from the die, and *V*_1_–*V*_6_ are the speeds of the rollers’ rotation.

**Figure 2 materials-16-00107-f002:**
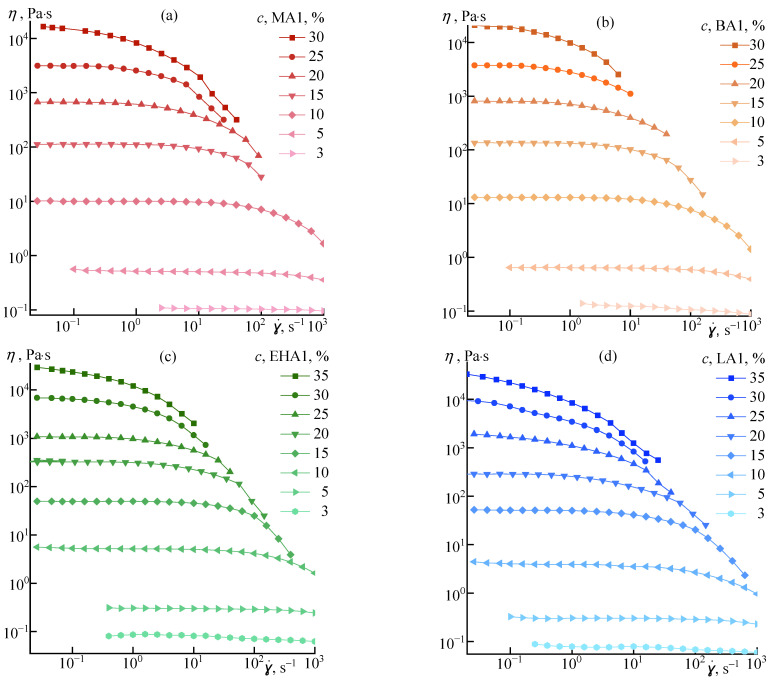
Flow curves of polymer solutions with *M_w_*~150 kg/mol (AlkA1 rank) of various concentrations at 25 °C, with various alkyl acrylates: (**a**)—methyl acrylate; (**b**)—butyl acrylate; (**c**)—2-ethylhexyl acrylate; (**d**)—lauryl acrylate. The concentrations are indicated in the legend.

**Figure 3 materials-16-00107-f003:**
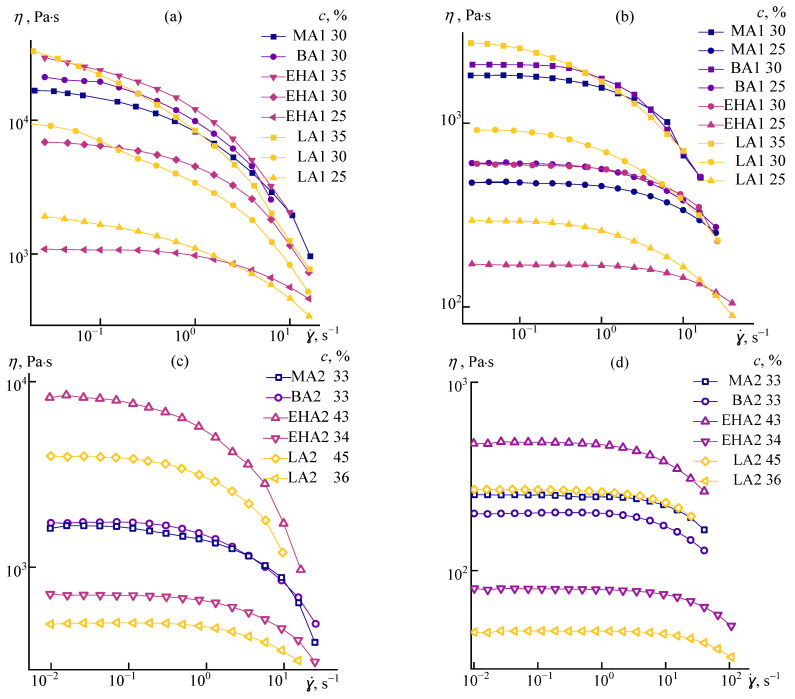
Flow curves of high molecular weight (AlkA1) and low molecular weight (AlkA2) concentrated solutions of different concentrations at 25 °C (**a**,**c**) and 70 °C (**b**,**d**). The concentrations are indicated in the legend.

**Figure 4 materials-16-00107-f004:**
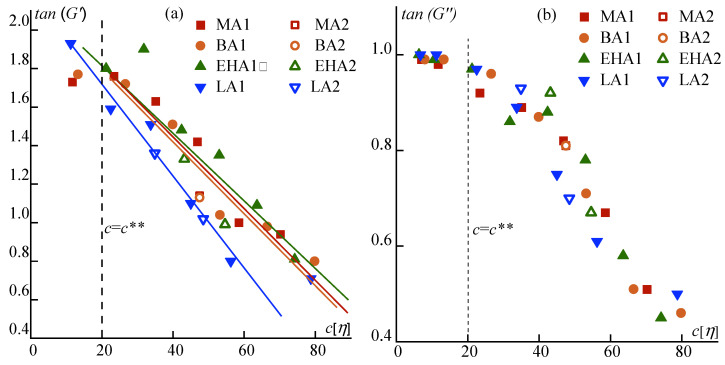
Dependences of the slope angle of the logarithmic frequency dependences of elastic (**a**) and loss (**b**) moduli in the terminal zone at 25 °C on the dimensionless parameter *c*[*η*] for PAN terpolymer solutions.

**Figure 5 materials-16-00107-f005:**
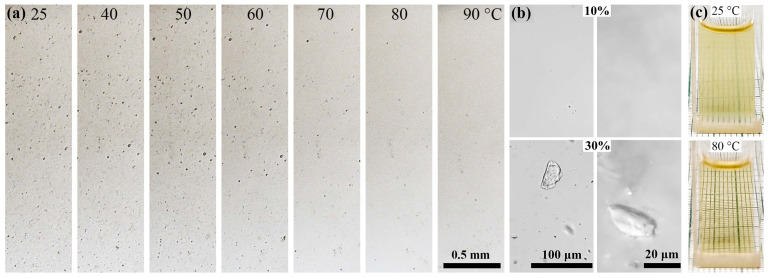
Changes in the morphology of a 35% PAN-LA1 solution in DMSO upon heating (**a**); images of PAN-LA1 solutions with the concentrations of 10% and 30% at 25 °C (**b**); and images of the 15% PAN-LA1 solution at 80 °C and 25 °C (4 h after heating) (**c**).

**Figure 6 materials-16-00107-f006:**
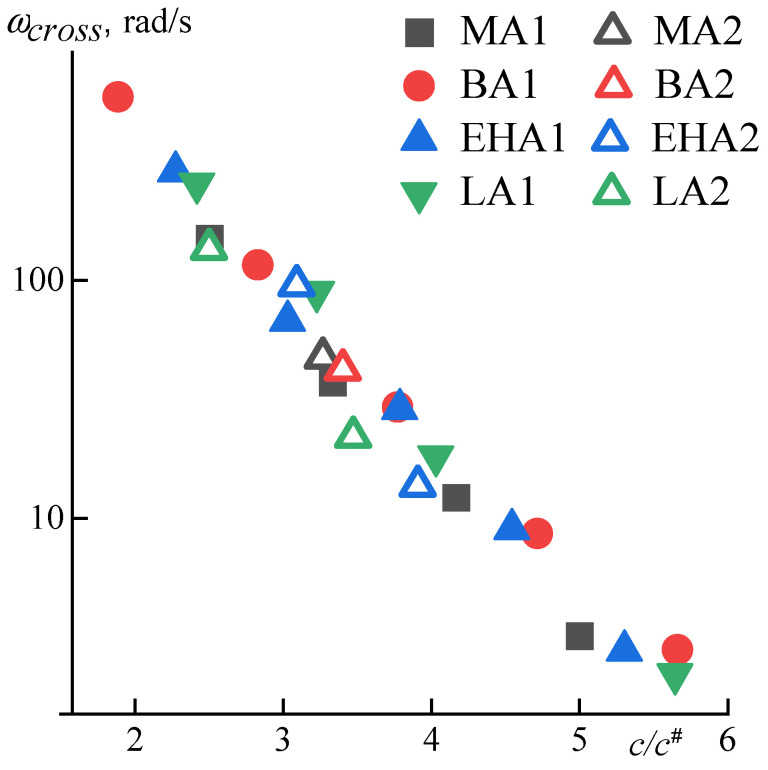
Dependence of the crossover frequency of terpolymers on the *c*/*c*^#^ parameter, which characterizes the entanglements’ density.

**Figure 7 materials-16-00107-f007:**
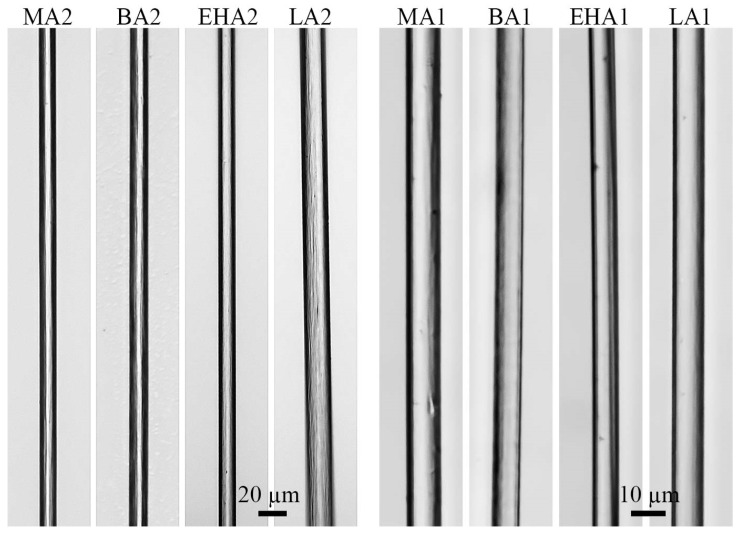
Optical micrographs of terpolymer-based fibers with various alkyl acrylate comonomers.

**Figure 8 materials-16-00107-f008:**
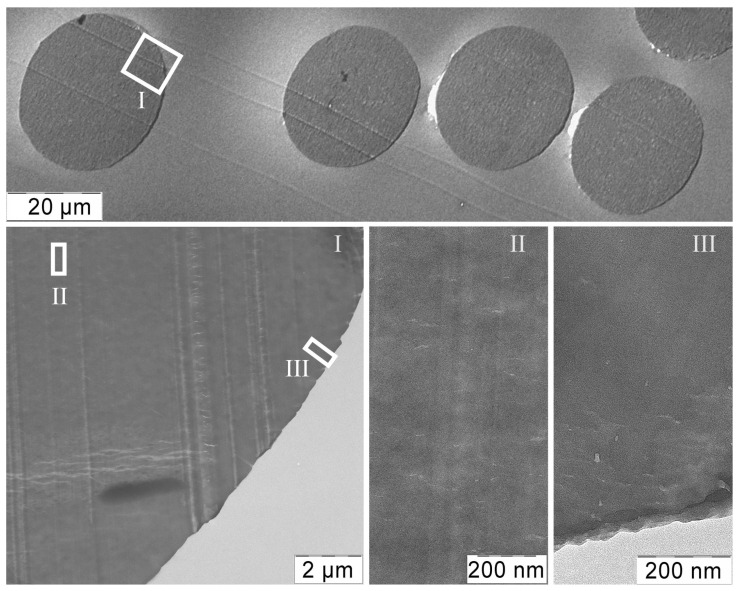
TEM images of PAN-MA1 fiber morphology (sample before orientation drawing). The arrow indicates the presence of a dense shell.

**Figure 9 materials-16-00107-f009:**
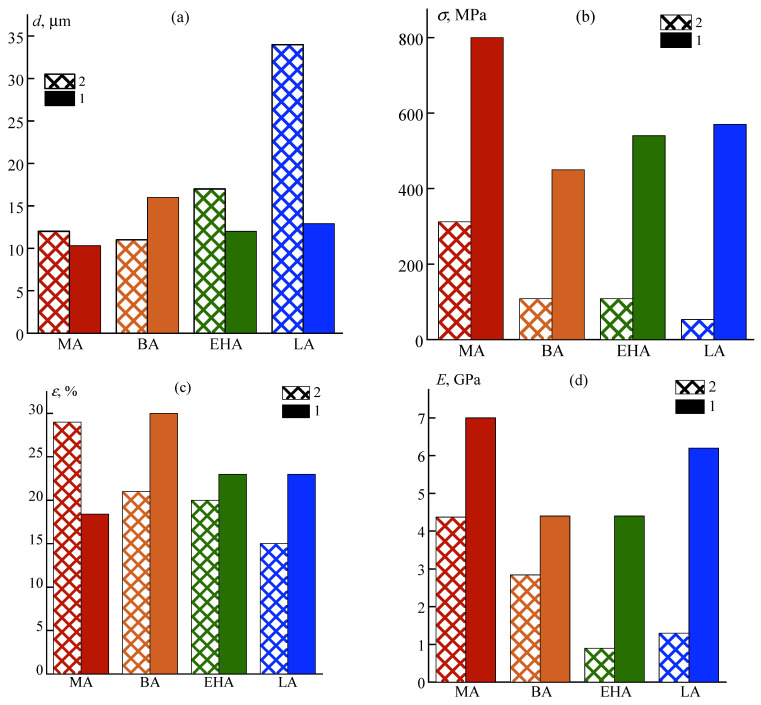
Mechanical characteristics of fibers made from copolymers with various alkyl acrylate comonomers: (**a**) diameter; (**b**) tensile strength; (**c**) elongation at break; (**d**) elastic modulus; 1—AlkA1 series with *M_w_*~150 kg/mol, 2—AlkA2 series with *M_w_*~50 kg/mol.

**Figure 10 materials-16-00107-f010:**
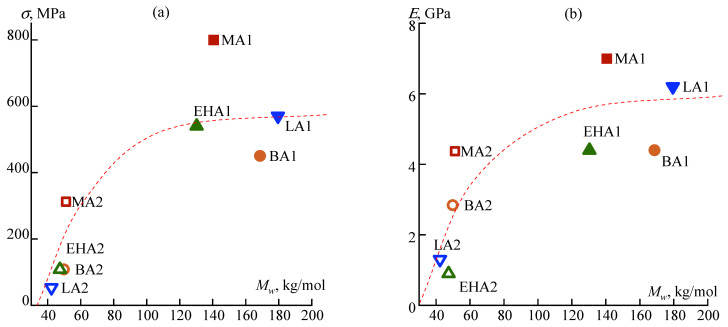
Correlation of strength (**a**) and elastic modulus (**b**) of fiber with a molecular weight of copolymers. The dotted line shows the expected saturation curves corresponding to the stabilization of the polymer properties.

**Figure 11 materials-16-00107-f011:**
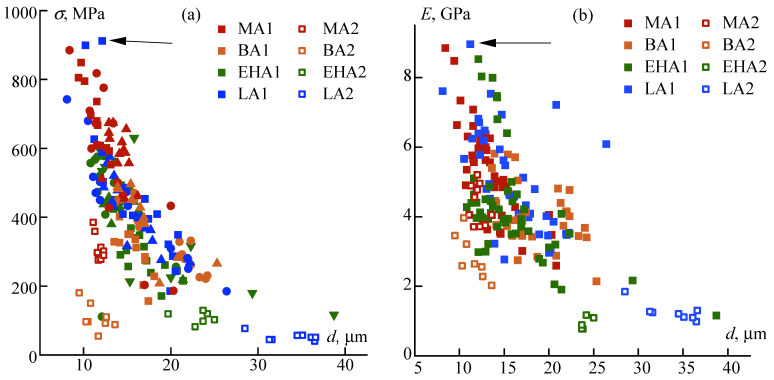
Dependences of strength (**a**); elastic modulus (**b**); and elongation at break (**c**) on fiber diameter.

**Table 1 materials-16-00107-t001:** Composition of terpolymers.

Designation	Alkyl Comonomer	*M_w_*, kg/mol	*Ð*	*F_AN_* *, mol%	*F_AlkA_* *, mol%	*F_AAm_* *, mol%
MA1	methyl acrylate	140.6	1.61	85.0	3.3	11.7
BA1	butyl acrylate	168.7	1.67	85.6	3.0	11.4
EHA1	2-ethylhexyl acrylate	130.5	1.61	85.2	3.0	11.8
LA1	lauryl acrylate	179.5	1.88	85.0	3.0	12.0
MA2	methyl acrylate	51.1	1.68	85.3	11.0	3.7
BA2	butyl acrylate	49.7	1.61	85.7	10.3	4.0
EHA2	2-ethylhexyl acrylate	47.4	1.71	84.5	11.0	4.5
LA2	lauryl acrylate	42.3	1.74	85.3	10.5	4.2

* where AN—acrylonitrile, AlkA—alkyl acrylate, AAm—acrylamide.

**Table 2 materials-16-00107-t002:** Viscometric characteristics of solutions of terpolymers in DMSO.

Designation	[*η*], dL/g	*k_H_*	*c**, g/dL	*c*^#^, g/dL	*c***, g/dL
MA1	2.34 ± 0.02	0.43 ± 0.08	0.3	6.0	47
BA1	2.66 ± 0.04	0.37 ± 0.13	0.3	5.3	53
EHA1	2.12 ± 0.02	0.38 ± 0.07	0.4	6.6	42
LA1	2.25 ± 0.05	0.35 ± 0.18	0.3	6.2	45
MA2	1.39 ± 0.03	0.37 ± 0.08	0.6	10.1	28
BA2	1.44 ± 0.01	0.39 ± 0.01	0.5	9.7	29
EHA2	1.27 ± 0.01	0.31 ± 0.03	0.6	11.0	25
LA2	0.97 ± 0.01	0.41 ± 0.03	0.8	14.4	19

*c**, *c*** are the estimated concentrations of the transition to the region of semi-dilute and concentrated solutions, and *c*^#^ is the concentration of the entanglement’s formation.

**Table 3 materials-16-00107-t003:** Spinning parameters for the terpolymers under investigation.

Sample	*c*, %	*t_s_*, °C	*V*_0_ * m/min	Rollers Velocity, m/min					Total Drawing Ratio (*V*_6_*/V*_1_)
				*V* _1_	*V* _2_	*V* _3_	*V* _4_	*V*_5_, *V*_6_	
MA1	30	70	0.08	4.1	9.8	14.5	16.0	22.8	5.6
BA1				4.2	11.4	12.5	13.3	14.7	3.5
EHA1	33		0.04	5.0	7.2	8.4	10.2	15.4	3.1
LA1	35			5.0	6.5	8.4	9.0	16.2	3.2
MA2	33	25	0.08	8.9	10.8	21.7	30.0	40.8	4.5
BA2				4.4	7.0	18.0	24.3	37.9	8.6
EHA2	43	30	0.04	3.5	4.2	8.7	12.0	17.0	4.8
LA2	45		0.08	4.4	5.0	6.5	6.9	15.0	3.4

* linear velocity of solution outflow from the spinneret.

## Data Availability

The data that support the findings of this study are available from the corresponding author upon reasonable request.

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
