# Peer review of "Influence of Alkyl Acrylate Nature on Rheological Properties of Polyacrylonitrile Terpolymers Solutions, Spinnability and Mechanical Characteristics of Fibers"

_materials, 2022, doi:10.3390/ma16010107_

Round 1

Reviewer 1 Report

In this manuscript, Skvortsov et.al synthesized a series of poly acrylonitrile copolymers with different alkyl acrylate comonomers through RAFT polymerization. The authors found these copolymers follow the same rheological behaviors, except for PAN-LA1, which exhibited poor solubility under characterization conditions. The authors further spun these polymers into fibers and characterized their mechanical properties of them. The authors found that the length of the alkyl chain of acrylates comonomers has a significant impact on the mechanical strength of the fibers. However, the discussion of the results is somewhat insufficient and confusing. I would reconsider recommending it for publication after major revision. Here are several comments below that would be helpful:

1.       In line 48, the definition of “structural method” is unclear. Did the authors mean the rheological method?

2.       In lines 58 – 61, it would be great if the authors can provide the references.

3.       In lines 74 – 77, “It is shown that the viscosity of their solutions is lower than for classical synthesis at the same MWD. It is assumed that the difference in the branching degree between AN and acrylamide copolymers leads to different thermal behavior of the copolymers at low temperatures.” The purpose of the latter sentence is unclear to me. Is the latter an explanation for the former?

4.       In line 83, It might be better to replace “the effect of their length” with “ the effect of their alkyl chain length”.

5.       In Table 1, the last row is the same as the first row.

6.       In lines 106 – 109, why did the authors define the entanglement concentration in this way? Can the author rationalize it? Is it a common definition in rheology? In polymer physics, the onset of entanglement defines regime III, and regime III should stand for the entangled regime instead of the concentrated regime. Namely, c** in figure 1 should be the entanglement concentration instead of c# by the definition in polymer physics. In the thermodynamic of polymer solution, when the correlation length of polymers becomes smaller than the thermal blob, the solution crossovers from the semi-dilute regime to the concentrated regime.

7.       The authors should increase the font size in figure 2.

8.       In lines 207 – 208, the meaning of “, which indicates a low degree of structuring of such solutions.” is unclear.

9.       In lines 223 – 225, how did the author define the terminal zone? Did the author do time-temperature superposition to reach the actual terminal zone? If not, by just calculating the slopes of G’ and G” in Figure A1, A2 definitely gives the results that differ from Maxwellian behavior, because some entangled chains have not yet finished their reptation.

10.   In lines 227 – 228, “while the elastic behavior of PAN-LA series solutions does not obey the general dependence.” How different it is compared to other copolymers? Did the authors do a statistical analysis?

11.   In figure 6, I can still see particles at 80 °C, how did the author determine the solution become homogeneous? Did the authors do dynamic light scattering?

12.   The discussion in lines 253 – 258 is confusing. In line 254, the solution should behave like a gel instead of an elastomer, and this is a consequence of entanglements that act like crosslinkers at certain timescales. People from different fields define mesh differently, using the term “the sleeping mesh network” will only make it more confusing. In addition, what is the meaning of “sleeping” in this term?

13.   In lines 308 – 309, besides the mechanical tests, is there any characterization (i.e. WAXS) to support the macromolecular orientation? Better mechanical properties are not direct evidence of better orientation.

14.   Why did the authors use weight average molecular weight in Figure 11 instead of number average molecular weight (in table 1)?

15.   The authors should explain more about the dashed line in Figure 11. Is it a model? Or just a guide to show the trend?

Author Response

Reviewer 1

In this manuscript, Skvortsov et.al synthesized a series of poly acrylonitrile copolymers with different alkyl acrylate comonomers through RAFT polymerization. The authors found these copolymers follow the same rheological behaviors, except for PAN-LA1, which exhibited poor solubility under characterization conditions. The authors further spun these polymers into fibers and characterized their mechanical properties of them. The authors found that the length of the alkyl chain of acrylates comonomers has a significant impact on the mechanical strength of the fibers. However, the discussion of the results is somewhat insufficient and confusing. I would reconsider recommending it for publication after major revision. Here are several comments below that would be helpful:

Thank you very much for taking the time to review our manuscript. We appreciate your detailed consideration of our manuscript. Many constructive suggestions which would help us to improve its quality are given. All changes are highlighted in the revised manuscript and the point-by-point responses to your comments are done. (Figures is in the attached file)

  1. In line 48, the definition of “structural method” is unclear. Did the authors mean the rheological method?

Not rheology, but we studied the solvent effect on the structure of solutions by the FTIR method [12], which made it possible to evaluate types of polymer–solvent and solvent–solvent interactions, as well as to evaluate slightly the effect of water, which very easily enters the aprotic solvents used for PAN dissolving. For better understanding, the phrase "structural method" has been replaced by “the FTIR method”.

  1. In lines 58 – 61, it would be great if the authors can provide the references.

We considered this as a well-known fact, but the link should be here, and it has been added.

  1. In lines 74 – 77, “It is shown that the viscosity of their solutions is lower than for classical synthesis at the same MWD. It is assumed that the difference in the branching degree between AN and acrylamide copolymers leads to different thermal behavior of the copolymers at low temperatures.” The purpose of the latter sentence is unclear to me. Is the latter an explanation for the former?

That sentence was rewritten. The difference between two types of synthesis is in their solution rheology, and thermal behavior.

  1. In line 83, It might be better to replace “the effect of their length” with “ the effect of their alkyl chain length”.

That's more accurate, the phrase has been corrected.

  1. In Table 1, the last row is the same as the first row.

The duplicate row has been removed

  1. In lines 106 – 109, why did the authors define the entanglement concentration in this way? Can the author rationalize it? Is it a common definition in rheology? In polymer physics, the onset of entanglement defines regime III, and regime III should stand for the entangled regime instead of the concentrated regime. Namely, c** in figure 1 should be the entanglement concentration instead of c#by the definition in polymer physics. In the thermodynamic of polymer solution, when the correlation length of polymers becomes smaller than the thermal blob, the solution crossovers from the semi-dilute regime to the concentrated regime.

The intrinsic viscosity characterizes the size of the polymer coil; accordingly, the c[h] indicates the volume occupied by polymer coils in solution. In dilute solutions, when there is no interaction and interpenetration between coils, the dependence of viscosity on volume concentration in log-log scales is linear with a slope of 1. A similar linear dependence is observed for concentrated solutions when a developed network of entanglements presents. In this case, the slope depending on the nature of the polymer/solvent mixture is 5-7. In our opinion, there is no difference between entangled and concentrated solutions. Traditionally, the most reproducible point is the intersection of these linear dependences, i.e. c# and density of entanglements network determines often as c/c#. But simultaneously it should be accepted that the region between C* and C** corresponds to the formation of a stable network of entanglements, and this region of deviation from linearity (between c* and c**) characterizes the zone of semi-dilute solutions with forming network of entanglements. This approach was developed in some publications (see below).

Figure R1. Viscosity concentration dependencies Fig5 Ebagninin, Fig 4 Graessley.

Simha, R. and Zakin, J.L., 1962. Solution viscosities of linear flexible high polymers. Journal of Colloid Science17(3), pp.270-287.

Graessley, W.W., Hazleton, R.L. and Lindeman, L.R., 1967. The Shear‐Rate Dependence of Viscosity in Concentrated Solutions of Narrow‐Distribution Polystyrene. Transactions of the Society of Rheology11(3), pp.267-285.

Clasen, C. and Kulicke, W.M., 2001. Determination of viscoelastic and rheo-optical material functions of water-soluble cellulose derivatives. Progress in polymer science26(9), pp.1839-1919.

Ebagninin, K.W., Benchabane, A. and Bekkour, K., 2009. Rheological characterization of poly (ethylene oxide) solutions of different molecular weights. Journal of colloid and interface science336(1), pp.360-367.

  1. The authors should increase the font size in figure 2.

Font size has been increased.

  1. In lines 207 – 208, the meaning of “, which indicates a low degree of structuring of such solutions.” is unclear.

The phrase has been rewritten to be more understandable in such a way:

It means a smooth change of viscosity with shear rates.

  1. In lines 223 – 225, how did the author define the terminal zone? Did the author do time-temperature superposition to reach the actual terminal zone? If not, by just calculating the slopes of G’ and G” in Figure A1, A2 definitely gives the results that differ from Maxwellian behavior, because some entangled chains have not yet finished their reptation.

Of course, in the limiting case of the relaxation spectrum, the dependences of the moduli should have slopes corresponding to the Maxwellian model. But for many real systems, especially gels, extremely high relaxation times are observed. Therefore, at reasonable observation times, the deviation from the model in the apparent "terminal" zone proceeds. For many systems, the principle of superposition is not very correct to be applied due to not traditional change in their behavior at different temperatures. Thus, the heating of PAN solutions leads to redistribution of intra- and intermolecular polymer-polymer and polymer-solvent interactions in concentrated solutions, and the principle of superposition is not acted (see Figure R2). Such modification of interactions leads either to change of their structure or even to form of emulsion. Below the data for concentrated solutions of copolymers with methacrylate and lauryl acrylate described in this article at two temperatures are shown. It is seen that for MA up to 25% content the superposition works well then deviations begin. For solutions of copolymers with LA deviations are observed for all systems because at 25°C it they transform to emulsions.

Fig. R2. Frequency dependences of dynamic moduli master curves for solutions of copolymers with MA- and LA-acrylates at 25 and 70oC.

Indeed, as the concentration increases, the angles become lower, that means increase in the relaxation times and deviations from the Maxwell model become stronger.

  1. In lines 227 – 228, “while the elastic behavior of PAN-LA series solutions does not obey the general dependence.” How different it is compared to other copolymers? Did the authors do a statistical analysis?

We always check the data for errors. Several solutions with the same concentration were prepared, and the flow curve and dynamic properties were measured at least 3 times for each concentration to check the reproducibility of the results. Moreover, we compared AN terpolymers with lauryl acrylate obtained from different syntheses with both AN-acrylamide, and AN-acrylic acid. For all these systems a similar behavior within the instrument error was found.

  1. In figure 6, I can still see particles at 80 °C, how did the author determine the solution become homogeneous? Did the authors do dynamic light scattering?

The homogeneity of the solutions was estimated by their transparency and the absence of drops of the second phase visible in an optical microscope. Indeed, the image shows several dust particles - which are present on the camera matrix and in the solution. Nevertheless, most of the drops dissolve when heated and the solution becomes transparent (see the figure R3 in the attachment). Due to the high viscosity and concentration of the polymer, the DLS method is not applicable for the correct measurement of purity and possible aggregates dimensions in such systems.

Fig. R3. Transparence of PAN-LA solution at 80 °C, and 25 °C (4 hours after heating)

  1. The discussion in lines 253 – 258 is confusing. In line 254, the solution should behave like a gel instead of an elastomer, and this is a consequence of entanglements that act like crosslinkers at certain timescales. People from different fields define mesh differently, using the term “the sleeping mesh network” will only make it more confusing. In addition, what is the meaning of “sleeping” in this term?

Thank you for your valuable comment. The phrase has been rewritten. We hope it looks clearer:

At high frequencies (above the crossover frequency), the solution behaves like a cross-linked gel, and its properties are determined solely by the density of the entanglements that have arisen in these conditions. The forerunners of them are fluctuating entanglements, which can be considered as a precursor of rubber-like nodes. As the mesh density increases, which can be characterized by the c/c# ratio, the crossover frequency decreases linearly in semilogarithmic coordinates.

  1. In lines 308 – 309, besides the mechanical tests, is there any characterization (i.e. WAXS) to support the macromolecular orientation? Better mechanical properties are not direct evidence of better orientation.

Unfortunately, the structure of these fibers wasn’t investigated, but our previous studies, as well as published data, indicate a clear correlation between the mechanical properties of PAN fibers and their orientation at close values of the polymer molecular weight. The phrase has been slightly changed.

  1. Why did the authors use weight average molecular weight in Figure 11 instead of number average molecular weight (in table 1)?

Molecular weight throughout the Manuscript has been changed to Mw for the text uniformity. 

  1. The authors should explain more about the dashed line in Figure 11. Is it a model? Or just a guide to show the trend?

The dotted line is just a common trend, typical for majority of polymer properties depending on molecular weight.

Reviewer 2 Report

The manuscript titled: “Influence of Alkyl Acrylate Nature of Rheological Properties of Polyacrylonitrile Terpolymers Solutions, Spinnability and Mechanical Characteristics of Fibers” is an interesting and well done paper.

1.  the main question addressed by the research:

The use of new terpolymers in base of different acrylates with acrylonitrile and acrylamide. The synthesis of these polymers was described previously, but in this manuscript they analysed the relevance of the comonomer in the processability, and in the final properties of the carbon fiber. 

2. whether the topic original or relevant in the field:

This is always difficult to say. The idea is not new, it is sure if you used different comonomers for PAN the final properties of the carbon fiber must be different. In my opinion, this manuscript is original because they use these copolymers of PAN for Carbon Fiber Production, doing a good study about the main properties they need for processing, i.e. rheology. Similar answer I have for the relevance, in my opinion CF are still a very relevant material in the industry with probably a large future, for that reason I think this paper in relevant, authors have presented a different methodology to obtain carbon fibers, then it is interesting for the research community in this area.

In my opinion this paper can be published as is, but before authors may remove the last row of table 1 because is the same as first one.

Author Response

Reviewer 2

Thank you very much for taking the time to review our manuscript. We appreciate your detailed consideration of our manuscript and thank you for appreciating our work.

The duplicate row has been removed.

Round 2

Reviewer 1 Report

The authors addressed most of the comments and the manuscript has been improved. I only have one last comment for this new version: I do not think there is no difference between an entangled solution and a concentrated solution. Fig 4 from Graessley shows exactly what I meant. The vertical line is the crossover from semi-dilute to concentrated, which is due to the correlation length becoming smaller than the thermal blob of the polymer so that the polymer chains have ideal chain statistics on all length scales. While the upper dashed/solid line is the crossover from the unentangled to the entangled regime. I am okay with the authors' definition as long as the authors clearly explain how they define the entanglement regime and the concentrated regime.